# Gas Biosensor Arrays Based on Single-Stranded DNA-Functionalized Single-Walled Carbon Nanotubes for the Detection of Volatile Organic Compound Biomarkers Released by Huanglongbing Disease-Infected Citrus Trees

**DOI:** 10.3390/s19214795

**Published:** 2019-11-04

**Authors:** Hui Wang, Pankaj Ramnani, Tung Pham, Claudia Chaves Villarreal, Xuejun Yu, Gang Liu, Ashok Mulchandani

**Affiliations:** 1Key Laboratory of Modern Precision Agriculture System Integration Research, Ministry of Education China Agri-Cultural University, Beijing 100083, China; wanghuilunwen@cau.edu.cn; 2Key Laboratory of Agricultural Information Acquisition Technology, Ministry of Agriculture China Agri-Cultural University, Beijing 100083, China; 3Department of Chemical and Environmental Engineering, University of California-Riverside, Riverside, CA 92521, USA; pramn002@ucr.edu (P.R.); tpham052@ucr.edu (T.P.); xyu010@ucr.edu (X.Y.); 4Materials Science and Engineering Program, University of California-Riverside, Riverside, CA 92521, USA; cchav021@ucr.edu; 5Research Institute of Wood Industry, Chinese Academy of Forestry, Beijing 100091, China

**Keywords:** ssDNA, SWNT, gas sensor, FET, volatile organic compounds, HLB

## Abstract

Volatile organic compounds (VOCs) released by plants are closely associated with plant metabolism and can serve as biomarkers for disease diagnosis. Huanglongbing (HLB), also known as citrus greening or yellow shoot disease, is a lethal threat to the multi-billion-dollar citrus industry. Early detection of HLB is vital for removal of susceptible citrus trees and containment of the disease. Gas sensors are applied to monitor the air quality or toxic gases owing to their low-cost fabrication, smooth operation, and possible miniaturization. Here, we report on the development, characterization, and application of electrical biosensor arrays based on single-walled carbon nanotubes (SWNTs) decorated with single-stranded DNA (ssDNA) for the detection of four VOCs—ethylhexanol, linalool, tetradecene, and phenylacetaldehyde—that serve as secondary biomarkers for detection of infected citrus trees during the asymptomatic stage. SWNTs were noncovalently functionalized with ssDNA using π–π interaction between the nucleotide and sidewall of SWNTs. The resulting ssDNA-SWNT hybrid structure and device properties were investigated using Raman spectroscopy, ultraviolet (UV) spectroscopy, and electrical measurements. To monitor changes in the four VOCs, gas biosensor arrays consisting of bare SWNTs before and after being decorated with different ssDNA were employed to determine the different concentrations of the four VOCs. The data was processed using principal component analysis (PCA) and neural net fitting (NNF).

## 1. Introduction

Citrus huanglongbing (HLB), also commonly called yellow shoot disease or citrus greening disease, is a lethal disease of citrus [1], which has become one of the most devastating diseases threatening citrus trees worldwide [2]. HLB-infected trees exhibit premature fruit drop, reduced fruit yield and quality, and eventually stop bearing fruits [3]. Production of citrus fruits is one of the most important agricultural activities in the world. According to the Food and Agriculture Organization, approximately 146 million tons of citrus fruits were produced in the year 2016, of which 7 million tons were produced in the United States, primarily in Florida and California. Due to the tremendous financial implications associated with HLB, early and rapid diagnostic techniques are necessary for effective mitigation of the disease.

HLB is associated with Gram-negative α-proteobacterium [4], such as *Candidatus* Liberibacter africanus, *Candidatus* Liberibacter americanus [5], and *Candidatus* Liberibacter asiaticus [1], which are transmitted through two psyllid species [6] and grafted seedling [7,8]. Although HLB was found in China in the 19th century [5], it was first reported in 1919 by Reinking [9]. Subsequently, the disease spread throughout Asia and Africa over the next decades, and then it took a toehold in Brazil. Today, HLB has spread all over the Americas, including the US. Suggested actions for mitigation of HLB include (i) planting of disease-free nursery stock, (ii) scouting for infected trees based on visual symptoms (yellow shoots, leaves with blotchy mottle, fruits with uneven coloration and reduced/lopsided shape) and subsequent removal, and (iii) control of psyllid vectors responsible for the spread of HLB by pesticide sprays. Early-stage detection of HLB is essential for identification and selective removal of infected trees, thereby preventing transmission of diseases to nearby healthy trees and orchids. However, due to the lengthy incubation and latent period of the pathogen, it has been shown that the infected trees can remain asymptomatic for extended periods of time [10,11]. Subsequently, visual symptoms, such as yellowing or asymmetrical yellow spots, start to display on parts of single shoot or branch [12], progressing to sparse yellow foliage, twig dieback, stunting, and ultimately reduced fruit yield [13,14] to half of their peak value. This renders early-stage diagnosis of HLB very challenging.

Current HLB diagnosis approaches include spectroscopic [12] and imaging-based techniques [15] and molecular techniques, such as enzyme-linked immunosorbent assay (ELISA) and reverse transcription polymerase chain reaction (RT-PCR)-based assays [16]. The visual identification methods are not suitable for diagnosis in the asymptomatic stage. Moreover, they are found to have low efficiency because the nonspecific nature of foliar symptoms makes the disease difficult to distinguish from nutrient deficiencies or other plant diseases. In contrast, ELISA and RT-PCR-based assays are highly sensitive and accurate. However, due to the low concentration and nonuniform distribution of bacterial cells (or DNA) in the tissues of infected trees, compounded by complex sample preparation and long assay times, these techniques are time-consuming, labor-intensive, and hence impractical for rapid detection of HLB that is required in case of an epidemic.

An alternative approach for early-stage detection of HLB is based on monitoring the contents of the volatile organic compounds (VOCs) emitted by the plant. Many researchers have demonstrated that the composition of VOCs emitted by plants can serve as an indicator of the status of a plant’s health and used to discriminate healthy and infected trees with high accuracy. Additionally, due to its noninvasive nature, it is amenable for in-field use compared to other methods [17]. In 2014, Aksenov and coauthors [10] published a paper illustrating that changes in the concentration of VOCs, including tetradecene, linalool, phenylacetaldehyde, and ethyl hexanol, were correlated with infected trees in the asymptomatic stage of HLB. The current state-of-the-art techniques for VOC detection typically utilize gas chromatograph (GC) in combination with various detectors, including flame ionization (GC-FID) [18], differential mobility spectrometry (GC-DMS) [19], or mass spectrometry (GC-MS) [20]. However, these instruments are expensive, require trained personnel to operate, and are not easily field-deployable due to their bulky size. Alternative quick and easy methods for VOC identification in environmental monitoring employ a target-specific chemical sensor or an electronic nose. 

Gas sensing technologies play a central role in monitoring the environment and the air quality in addition to detecting toxic gases. Gas sensors of various types have been employed, but the most popular ones are resistivity-based sensors owing to their low-cost fabrication, smooth operation, and possible miniaturization. A primary challenge is to design and develop robust, reliable gas sensors that are highly sensitive and selective to target gases, which can be operated at or close to room temperature under the influences of humidity [21].

Single-walled carbon nanotubes (SWNTs), due to their high surface area to volume ratio and semiconducting nature, are regarded as an ideal transducer material for sensing applications [22,23]. Even though SWNT-based sensors have high sensitivity for gas molecule, one drawback is their lack of selectivity. To improve sensitivity and selectivity, biosensors that incorporate a variety of bioreceptors and transducers have been reported in the literature. SWNT surface can be functionalized/decorated with inorganic molecules (TiO_2_, SnO_2_ nanoparticles) [24,25,26], organic macromolecules (porphyrin, polycyclic aromatic hydrocarbons) [27,28], or biomolecules [29] to improve the sensing performance. DNA molecules [30,31] are naturally occurring polymers that have many unique functions, including catalyzing chemical reactions [32] and controlling gene expression. Compared to synthetic polymers, peptides and lipids, DNA offers several advantages. For example, they have (i) more precise composition compared to synthetic polymers, (ii) higher degradation stability compared to polypeptides and RNA, and (iii) greater structural and compositional diversity than lipids. Furthermore, the complex 3D structure attained by DNA oligomers may modify intermolecular interactions between DNA and vapor molecules, enhancing chemical sensing specificity. Recently, single-stranded DNA (ssDNA)-modified carbon nanotubes were demonstrated to sense and discriminate highly similar VOCs, including structural isomers and enantiomers, with excellent performance [33].

In this work, we propose a novel biosensor array using SWNTs functionalized with ssDNA through π-stacking, which improved the sensitivity of bare SWNTs. Two DNA oligomers previously applied for successful identification and detection of ppm to ppb levels of VOCs were used [33]. The modified biosensor was characterized via different methods, such as Raman spectroscopy, ultraviolet (UV) spectroscopy, and electrical measurements. The prepared biosensor array was applied to determine four VOCs (ethylhexanol, linalool, tetradecene, and phenylacetaldehyde), the emissions of which are altered by infected citrus tree at the asymptomatic stage. To obtain the changes in VOCs, different mathematical models were employed to distinguish the species and contents of VOCs.

## 2. Materials and Methods 

### 2.1. Chemicals and Materials 

Ninety-five percent semiconducting SWNT solution with a concentration of 0.01 mg/mL was purchased from Nano-Integris Inc. (Skokie, IL, USA). Acetone, propanol, and ammonium hydroxide were acquired from Fisher Scientific Company (Pittsburgh, PA, USA). 3-Aminopropyltriethoxysilane (APTES), 2-ethyl-1-hexanol, 1-tetradence, and phenylacetaldehyde were purchased from Sigma-Aldrich (Saint Louis, MO, USA). Linalool (97%) was obtained from Acros Organics. Milli-Q water was used throughout the experiments. The ssDNA were bought from Integrated DNA Technologies, Inc. (Coralville, IA, USA):
s1-DNA: 5′-CTT CTG TCT TGA TGT TTG TCA AAC-3′ and s2-DNA: 5′-AAA ACC CCC GGG GTT TTT TTT TTT-3′.

They were prepared with Milli-Q water and stored at −4 °C before use. 

### 2.2. Apparatus

The SWNTs before and after being noncovalently functionalized with ssDNA were characterized by Raman spectroscopy and UV–visible spectroscopy. Raman spectra were collected using Dilor XY Laser Raman with imaging microscope (514 nm diode and Ar ion lasers). UV spectra were recorded on a Varian1E spectrophotometer. A semiconductor parameter analyzer (Keithley 2636) was used to measure current–voltage (I–V) and field-effect transistor (FET) to confirm the successful modification of SWNTs with different materials and elucidate the sensing mechanism. The rest of the electrochemical measurements were collected using a gas sensing setup, which was designed and integrated by our group. 

### 2.3. Fabrication of Biosensor Arrays

SWNT-based biosensor arrays were fabricated using the protocol reported by Ramnani et al. [34]. Briefly, single gap microelectrodes were written on the SiO_2_/Si substrate using standard lithographic patterning, followed by deposition of a 20 nm Cr layer and a 180 nm Au layer successively via e-beam evaporation. The gap of microelectrode was designed to be 10 μm wide by 10 μm long. Figure 1 shows a schematic of the device fabrication steps. The bare electrodes were rinsed sequentially with acetone and isopropanol, cleaned in ammonia, and dried in air prior to use. They were used to remove the organic and inorganic substances on the surface. The electrodes were then incubated in 0.5 mL APTES for 60 min, washed with Milli-Q water, and blow-dried with nitrogen. 

SWNT solution was dispensed over the microelectrode surface and incubated in dark for 60 min under high humidity conditions to prevent fast evaporation. The residual SWNTs were washed, and the SWNT-functionalized electrodes were annealed in ambient air at 250 °C for 60 min. Finally, SWNTs were functionalized with ssDNA strands by applying a 10 μL ssDNA solution on the device for 4 h under high humidity conditions, cleaned with Milli-Q water to remove the excess ssDNA, and dried in ambient air.

### 2.4. Gas Sensing Setup

The gas sensing setup [35], shown in Figure 2, was divided into two parts—gas delivery and data collection—which was controlled by a software program using LabView. For the controlled gas delivery to the biosensor arrays, an air cylinder was connected to the inlet ports of two mass flow controllers (MFCs). One MFC was employed to control the flow rate of dry air through a bubbler unit containing the VOC, which could generate saturated vapors of the VOC, while the other MFC was used to control the flow rate of dry air. The ratio between saturated vapor and dry air was controlled by varying the MFCs’ flow rates to achieve certain concentration. For real-time data collection, the electrodes were connected to Keithley 2636 sourcemeter using a clip, and 0.1 V was applied between the drain and the source without the gate voltage. The electrode was covered with a 1.2 cm^3^ sealed glass dome, which had inlet and outlet ports. The biosensor arrays were first exposed to dry air until a stable baseline with the flow rate of 50 sccm, followed by exposure to analyte for 20 min and switching back to dry air for 30 min.

## 3. Results

### 3.1. Verification of ssDNA-SWNT Hybrid Formation

Raman spectroscopy is a powerful method for the characterization of carbon-based materials. It can be seen from Figure 3 that the Raman spectrum of s2DNA-SWNT showed the G^+^ band to be downshifted by 8 cm^−1^, and the width of the G^+^ band was reduced relative to the SWNTs. This shift in the G^+^ band can be attributed to charge transfer between the s2DNA and SWNTs. Another explanation for this downshift may be a compressive stress induced in the SWNTs because of the ssDNA wrapping on the nanomaterial [36]. Additionally, the shift in G band can also be caused by heavily functionalized nanotubes. The relative intensity of G^−^ to G^+^ band was reduced 9% relative to bare SWNTs [37]. The G^+^ and G^−^ bands can be predominantly ascribed to the vibrations of carbon atoms along the axis and the circumferential direction of the SWNT, respectively [38].

For the UV–vis spectra collection of SWNTs before and after functionalization with ssDNA, quartz plate was chosen as the substrate. As shown in Figure 4, the background absorbance spectrum of bare quartz glass did not reveal any absorption in the entire 200–800 nm wavelength region. Bare SWNTs immobilized on quartz exhibited an absorbance peak at 260 nm and a shoulder at 460 nm. The former was attributed to the π-plasmon resonance absorption and the latter to van Hove transition of metallic SWNTs (M_11_) [39]. Upon functionalization with s2DNA, there was a sharp increase in the absorbance peak at 260 nm. This increase was ascribed to the characteristic ssDNA peak at 260 nm [40], confirming that attachment of ssDNA to SWNTs generated the composite effect [41].

The device fabrication steps, including SWNT immobilization and functionalization with ssDNA, were characterized by recording the current–voltage (*I_DS_–V_DS_*) characteristics after each step of surface modification. Figure 5a shows the *I_DS_–V_DS_* characteristics of bare SWNTs and s2DNA-modified SWNTs. The resistance (determined from the inverse of the slope of *I_DS_–V_DS_* curve between +0.1 and −0.1 V) of the SWNTs increased ~5-fold, from 15 to 73 kΩ, when modified with ssDNA. The large increase was attributed to the accumulation of negative charges from the phosphate backbone of DNA [42]. While the resistance of SWNTs increased upon modification with s1DNA (Appendix A), the increase was smaller (from 1.47 to 4.56 kΩ) than for s2DNA functionalization even though both s1DNA and s2DNA had identical number of bases. This can be attributed to the difference in the ssDNA sequences and type of bases that resulted in different heterostructure of the ssDNA-SWNT hybrid. 

The functionalization of SWNTs with ssDNA was further confirmed by obtaining FET transfer characteristics of SWNTs before and after s2DNA functionalization. As shown in Figure 5b, the threshold gate voltage (*V_TH_*) of bare SWNT was −9.6 V. Compared with the bare SWNTs, the transfer curve for s2DNA-SWNT shifted to the negative direction and had a more negative threshold gate voltage of −26 V. Additionally, upon functionalization with s2DNA, the mobility decreased from 168.8 cm^2^/Vs for bare SWNTs to 121 cm^2^/Vs for s2DNA-SWNTs. The shift in threshold voltage and decrease in hole mobility can be explained by the reduction in hole concentration of p-type SWNTs due to charge (electron) transfer from negatively charged phosphate backbone of s2DNA. 

### 3.2. Sensing of VOCs Specific to HLB

As discussed earlier, VOCs, including phenylacetaldehyde, tetradecene, linalool, and ethylhexanol, can serve as biomarkers for detection of HLB-infected citrus trees in their asymptomatic stage. To evaluate the performance of ssDNA-functionalized SWNT devices, we measured the real-time electrical response of bare SWNT, s1DNA-SWNT, and s2DNA-SWNT chemiresistor biosensors to phenylacetaldehyde, tetradecene, linalool, and ethylhexanol vapors with concentration varying from 5% to 100% of saturated vapors at room temperature.

Figure 6a shows the dynamic response of normalized resistance change (ΔR/R_0_% = (R − R_0_)/R_0_ × 100%, where R_0_ is the initial baseline resistance in air, and R is the resistance in different concentration of VOCs) at *V_DS_* = 0.1 V and base voltage (*V_G_*) = 0 V. Based on calibration plots shown in Figure 6b, it was found that three devices—bare SWNTs, s1DNA-SWNTs, and s2DNA-SWNTs—were able to detect phenylacetaldehyde vapors over a large concentration window of 5–100% of saturation. The data showed that all the biosensors in the array exhibited a nonselective response to phenylacetaldehyde, with bare SWNT and s2DNA-SWNT devices showing an increase in resistance, whereas the s1DNA-SWNT device exhibited a decrease in resistance. Furthermore, the sensitivity of the three sensors was in the following order: s2DNA-SWNTs > s1DNA-SWNTs > bare SWNTs. The best sensing characteristics were observed using s2DNA-SWNTs, with a sensitivity of 5.97% and a linear response (*R*^2^ = 0.99) within the tested window of concentration. Additionally, the recovery response time upon exposure to dry air was of the order of a few minutes, which makes the proposed gas biosensors very viable for real-time, on-site operations. 

The response of the three chemiresistive sensors—bare SWNT, s1DNA-SWNT, and s2DNA-SWNT—was not limited to phenylacetaldehyde, as evident from the calibration plot (Figure 7) for the remaining three VOCs (ethylhexanol, tetradecene, and linalool), which are known biomarkers of HLB-infected trees in their asymptomatic phase. Details of calibration plots for ethylhexanol, tetradecene, and linalool at varying concentrations from 5% to 100% are presented in Appendix A. The sign and magnitude of ssDNA-SWNT responses depended on both the DNA sequence used and the analyte. Nicholas and coworkers [29] proposed a possible reason why different ssDNA decorated with SWNTs would result in different complex, sequence-specific set of binding pockets located within a few nanometers of the SWNT sidewall. Different VOC molecules were solvated by the DNA hydration layer and then bound in the pockets, resulting in different ssDNA-SWNT responses.

### 3.3. Reproducibility

To study the reproducibility, four s2DNA-SWNTs were fabricated following the protocol in Section 2.3. They were exposed to different concentrations of phenylacetaldehyde varying from 5% to 100% of saturated vapors at room temperature, as shown in Figure 8. The errors of relative resistance at different concentrations were lower than 10%, confirming that the as-prepared s2DNA-SWNTs had high reproducibility.

### 3.4. Sensing Mechanism

In order to experimentally identify the sensing mechanism, the ssDNA-SWNT chemiresistor devices were operated in a field-effect transistor mode using Si as the back-gate electrode. The changes in the transfer characteristic curves (*I_DS_–V_G_*_,_
*V_DS_* = 0.1 V) upon functionalization with ssDNA and exposure to the VOCs was used to explain the electrostatic interactions of the DNA and VOC molecules with the SWNTs.

Figure 9a shows the transfer characteristic (*I_DS_–V_G_*) curves for s1DNA-SWNTs exposed to air and saturated phenylacetaldehyde vapors. As illustrated in the figure, transconductance (slope of the transfer characteristic curve) changed after exposure to saturated phenylacetaldehyde vapors. The increase in device transconductance upon exposure to analyte was attributed to change in local work function of the metal contacts and band alignment when the VOC molecules were adsorbed on the metal (gold) electrodes. When the s2DNA-SWNT device was exposed to phenylacetaldehyde vapors (), the *I_DS_–V_G_* curve in Figure 9b shifted to the negative direction, and *V_TH_* also decreased. This negative shift of *V_TH_* was attributed to adsorption of partially charged/polar VOC molecules, which induced a screening charge (doping) of the SWNTs and shifted the *I_DS_–V_G_* curve to a negative voltage. This mechanism, known as electrostatic gating, is commonly seen in most SWNT-based sensing devices. 

The type and number of bases in s1DNA and s2DNA are different, which caused the different properties of s1DNA-SWNTs and s2DNA-SWNTs.

### 3.5. Chemometric Analysis 

To monitor the changes of VOCs, the data collected from the gas biosensor array was processed by principal component analysis (PCA) and neural net fitting (NNF). 

#### 3.5.1. Principal Component Analysis

PCA is a data analysis tool used to reduce the dimensionality of a large number of interrelated variables while retaining the variance of the data set. A data matrix was constructed consisting of three columns representing peak responses of three biosensor (bare SWNT, s1DNA-SWNT, s2DNA-SWNT) arrays and the rows representing the four VOCs at seven concentrations (5%, 10%, 20%, 40%, 60%, 80%, and 100% saturation). As shown in Figure 10, the biosensor array of the three devices could unambiguously classify the four VOCs with concentrations ranging from 40% to 100%. It can be observed that the scores representing each gas are clustered together, and there is a clear separation between the clusters corresponding to each gas. Principal component 1 (PC1), represented on the x-axis, contained information about the concentration of the three analytes (linalool, tetradecene, and phenylacetaldehyde). The concentration of these analytes increased as the magnitude of PC1 increased and followed a definite trend between different concentrations of analytes. Meanwhile, principal component 2 (PC2), represented on the y-axis, provided information about the concentration of ethylhexanol and the discriminatory power of the sensors. However, for lower concentrations of VOC (Appendix A), it was hard to classify the response of the four VOCs based on the biosensor array. Classification at lower concentration is achievable by expanding the number of biosensors in the array using additional ssDNA sequence-functionalized devices. Alternatively, a sample preconcentration step upstream of the array can be employed to realize the goal.

We further investigated the response of the biosensor array to water vapor, and the results are shown in Appendix A. When the response for different water vapor concentrations was included in the PCA analysis (Appendix A), we found that the scores representing each gas overlapped, except for water vapor. Based on these results, it is recommended that, for real-world applications, water vapor be removed during sampling of the VOCs from the citrus trees.

#### 3.5.2. Neural Net Fitting

The input matrix (*X*_0_) consisted of the peak responses of the gas biosensors (bare SWNTs, s1DNA-SWNTs, s2DNA-SWNTs) at different concentrations of VOCs, and the output matrix (*Y*_0_) was different concentrations of the four VOCs and water (5%, 10%, 20%, 40%, 60%, 80%, and 100% saturated vapor). The input matrix (*X*_0_) was pretreated as follows:$$X=\left[1,\;x_{1},\;x_{2},\;x_{3},\;x_{1}^{2},\;x_{2}^{2},\;x_{3}^{2},\;x_{1}*x_{2},\;x_{1}*x_{3},\;x_{2}*x_{3}\right]$$
where $x_{1}$, $x_{2}$, and $x_{3}$ are the relative resistance of bare SWNTs, s1DNA-SWNTs, and s2DNA-SWNTs, respectively.

Neural net fitting was used to build the prediction model. *X*_0_ worked as the input layer, consisting of 36 columns and 10 rows using different concentrations of the four VOCs and water. *Y*_0_ was selected as the output layer, consisting of 36 columns and 5 rows. The data of *X*_0_ was divided randomly into three sets: 70% as training samples, 15% as validation samples, and 15% as testing samples. After the model was established, the predicted values from the NNF were compared to the true values, which are exhibited in Figure 11.

The correlation coefficient (*R*_0_) and root mean square error (*R_SMT_*) between the real and predicted concentrations were 0.984 and 4.04, respectively. 

## 4. Conclusions

Electrical sensor arrays were fabricated using SWNTs decorated with ssDNA with a facile drop-casting method. They were used to measure ethylhexanol, phenylacetaldehyde, linalool, and tetradecene, which are released by HLB-infected citrus trees in the asymptomatic phase of the disease. The ssDNA-SWNT devices showed a significant improvement in sensitivity compared to bare SWNTs, which also had excellent recovery and reproducibility in the validated range of concentration of analytes. In addition, the electrical sensor arrays were able to distinguish VOCs over a wide concentration range of 5% to 100% using NNF. In the simulated experiment, the VOCs could be successfully distinguished, but it was still difficult to apply in the field due to the poor specificity. The mixture of VOCs emitted by infected citrus trees is the next key research focus for our group. This can render potent platforms with high specificity for real-time, on-field detection of VOCs and thus aid in early-stage detection of trees infected with HLB and other diseases.

## Figures and Tables

**Figure 1 sensors-19-04795-f001:**
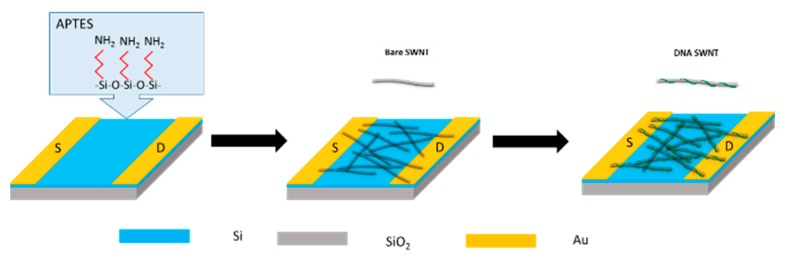
Schematic of single-walled carbon nanotube (SWNT)-wrapped DNA biosensor fabrication.

**Figure 2 sensors-19-04795-f002:**
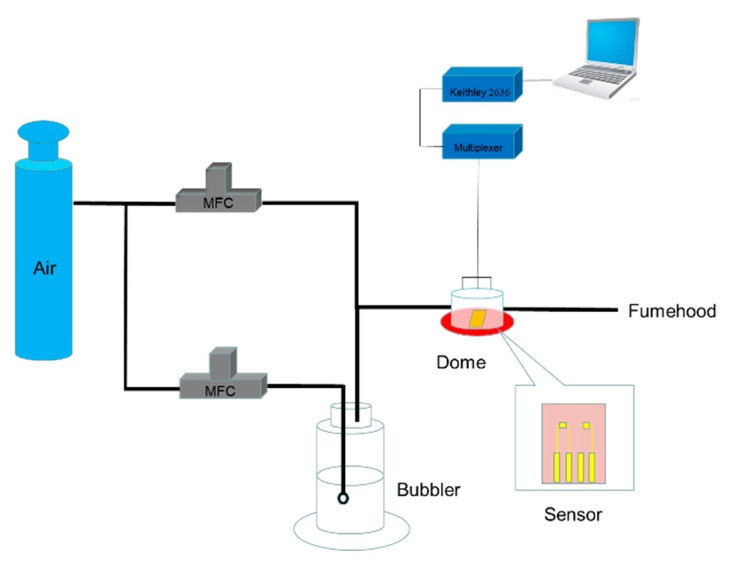
Schematic diagram of gas sensing device.

**Figure 3 sensors-19-04795-f003:**
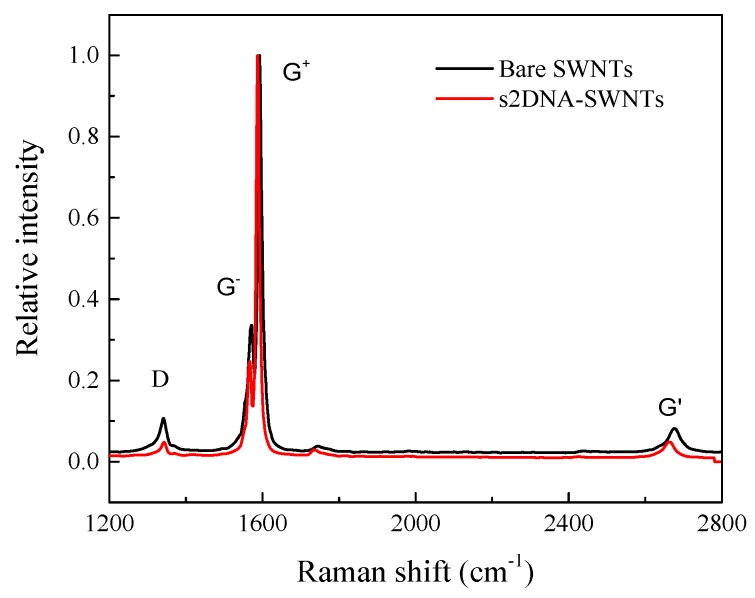
Raman spectra for bare SWNTs (blank) and s2DNA-wrapped SWNTs.

**Figure 4 sensors-19-04795-f004:**
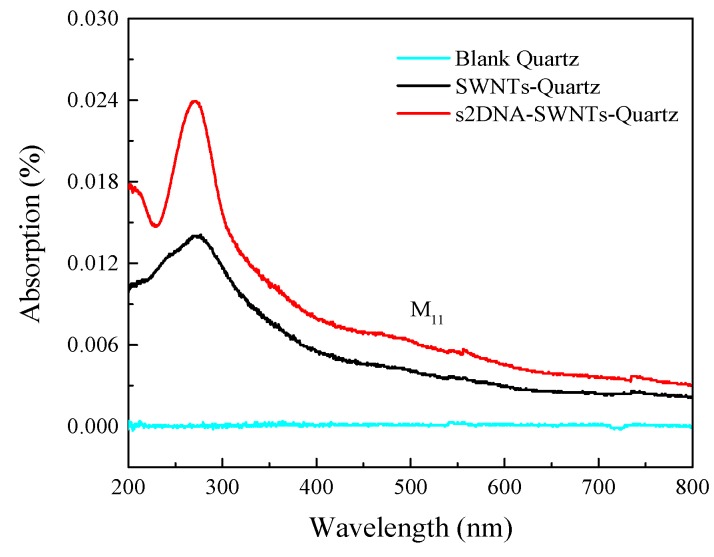
UV–vis absorption spectra of blank quartz (blue), SWNT–quartz (black), and s2DNA-SWNT–quartz (red).

**Figure 5 sensors-19-04795-f005:**
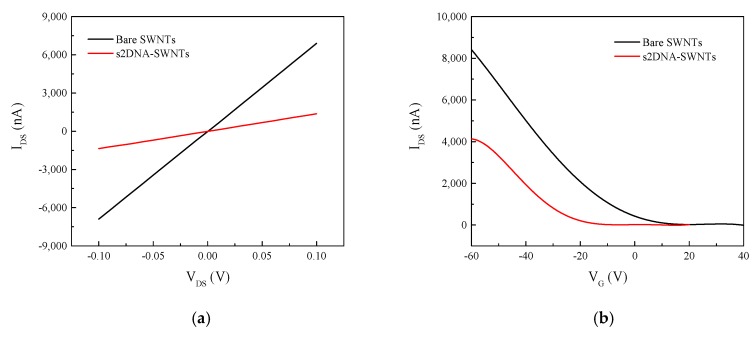
(**a**) Drain-source Current–voltage (*I_DS_–V_DS_*) characteristics of SWNTs before and after functionalization with s2DNA at base voltage (*V_G_*) = 0 V. (**b**) Transfer characteristic curve for bare and s2DNA-coated SWNT device at *V_DS_* = 0.1 V.

**Figure 6 sensors-19-04795-f006:**
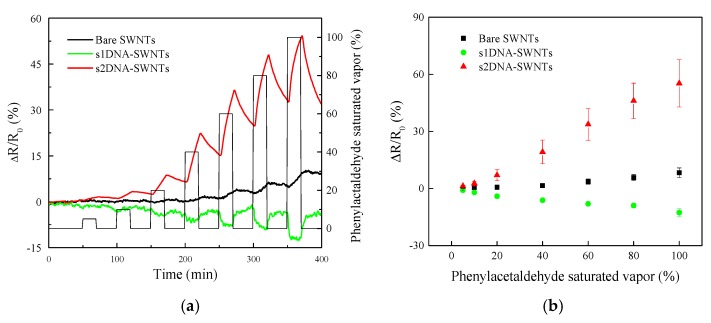
(**a**) Dynamic responses of bare and single-stranded DNA (ssDNA)-coated SWNT sensors toward different concentrations of phenylacetaldehyde vapors performed at *V_DS_* = 0.1 V. (**b**) Calibration curves for bare and ssDNA-coated SWNT devices. Each data point is an average of measurements from four independent biosensors.

**Figure 7 sensors-19-04795-f007:**
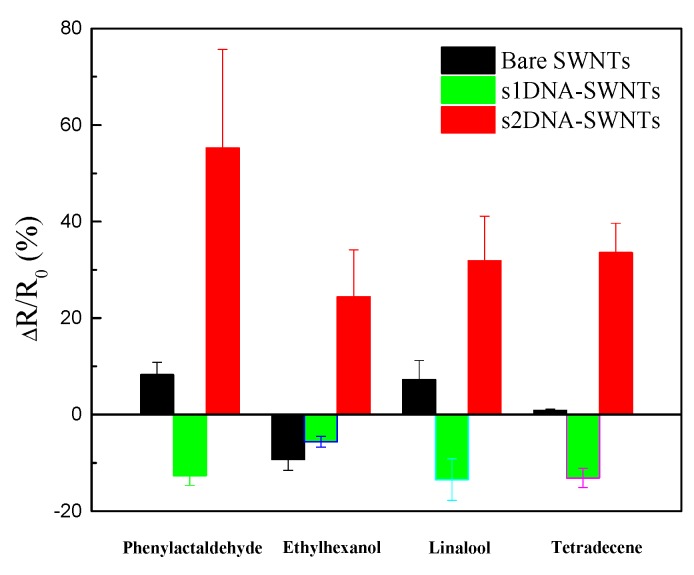
Histogram showing comparison of responses of bare SWNTs before and after being functionalized with ssDNA toward phenylacetaldehyde, ethylhexanol, linalool, and tetradecene (100% saturated vapors). Each data point is an average of measurements from four independent biosensors.

**Figure 8 sensors-19-04795-f008:**
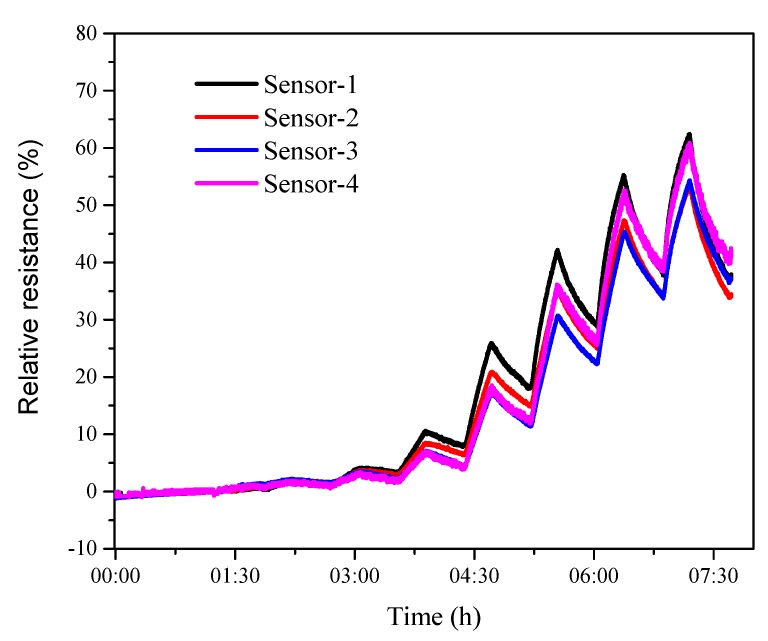
Dynamic responses of different s2DNA-SWNTs toward different concentrations of phenylacetaldehyde vapors performed at *V_DS_* = 0.1 V.

**Figure 9 sensors-19-04795-f009:**
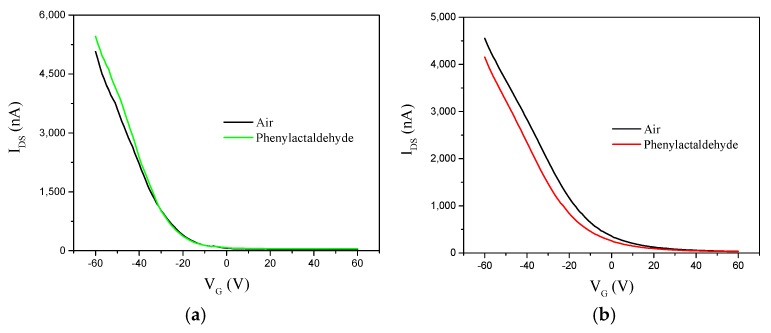
Transfer characteristic (*I_DS_–V_G_* curves at *V_DS_* = 0.1 V) of (**a**) s1DNA-SWNT and (**b**) s2DNA-SWNT devices in presence of air and saturated phenylacetaldehyde.

**Figure 10 sensors-19-04795-f010:**
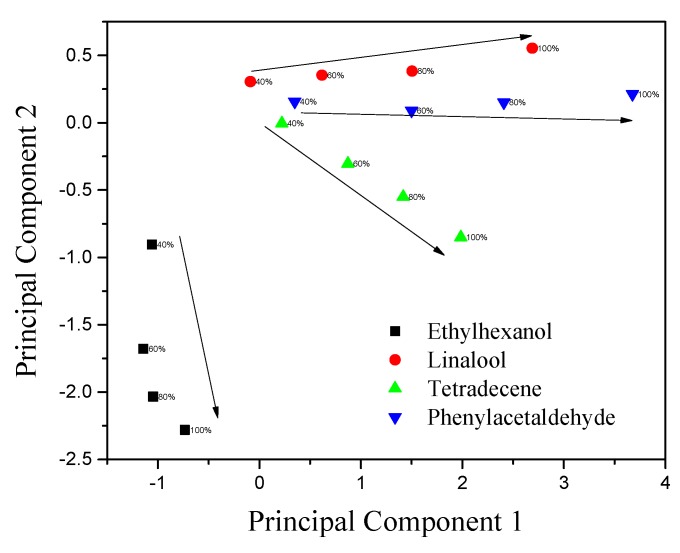
Principal component analysis (PCA) plot (PC1 vs. PC2) of scores using three sensors (bare SWNTs, s1DNA-SWNTs, and s2DNA-SWNTs) showing well-separated clusters for the four volatile organic compounds (VOCs) tested.

**Figure 11 sensors-19-04795-f011:**
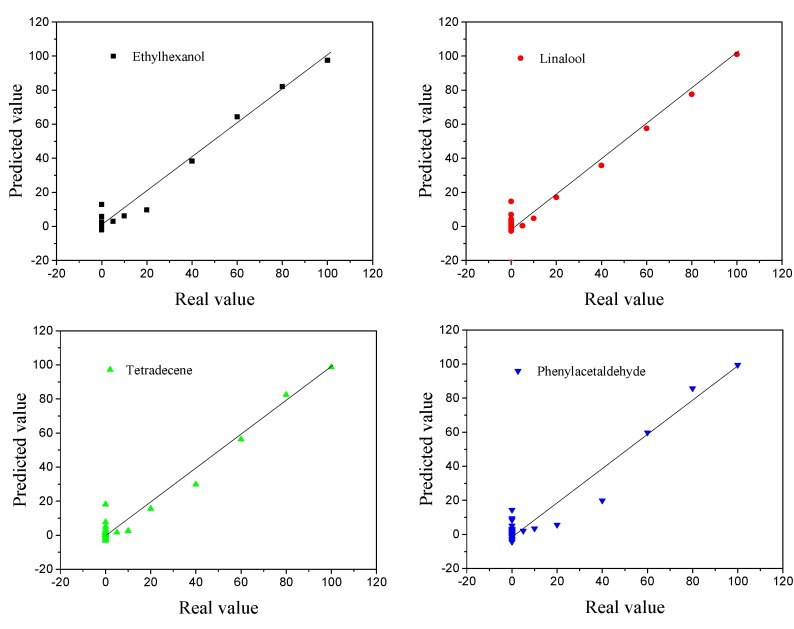
The real value versus the predicted value toward the different concentrations of four VOCs (ethylhexanol, linalool, tetradecene, and phenylacetaldehyde) calculated by neural net fitting (NNF).

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
