# Peer review of "Gas Biosensor Arrays Based on Single-Stranded DNA-Functionalized Single-Walled Carbon Nanotubes for the Detection of Volatile Organic Compound Biomarkers Released by Huanglongbing Disease-Infected Citrus Trees"

_sensors, 2019, doi:10.3390/s19214795_

Round 1
Reviewer 1 Report
Suggestions for Authors
Authors aimed at assessing the potential of SSDNA functionalized single walled carbon nano-tubes gas biosensor arrays for early detection of citrus trees infected with Huanglongbing disease. The subject is relevant for the maintenance of the yield of citrus. This manuscript is generally well written, proofreading is however required, since there are some minor grammatical revisions needed. I have outlined some suggestions for authors below.
Introduction
I would like to advise authors to make some adjustments to the structure of the introduction. Authors start by discussing the pitfalls of VOC analytical devices, however, the aim of the authors is to assess the ability of a specific eNose device for early detection of infected trees with HLB. I would like to advice authors to start by explaining the current situation regarding HLB and the need for novel detection methods. Thus, starting off with line 57 and the first two paragraphs can be transferred to line 92. This would increase reader comprehensibility, since all information regarding VOC analyses is then grouped together.
Line 45: “A primary challenge for gas sensors is therefore….” Here, “for gas sensors is therefore”, can cursiefbe removed from the sentence.
Line 45: Could authors provide an appropriate reference for this statement?
Line 48: Reference 1 should be placed at the end of the sentence.
Line 51: “bio/sensors” should be replaced by “bio-senors”
Line 52: “Bio/receptors” should be replaced by “bio-receptors”
Line 58: “Threathing” should be replaced by “threatening”.
Line 60: Could authors provide an appropriate reference for this statement?
Line 66: names of bacteria should be displayed in italic. Please revise through whole manuscript.
Line 99: Between linalool and phenylacetaldehyde a comma is missing.
Line 99: The names of the compounds should be checked for correct spelling.
Line 112: “was” should be adjusted to “were”.
Materials and methods
Lines 132-134: did authors publish these gas sensing setup elsewhere? If so, please provide a reference. If not, could authors provide the precise settings.
Line 136: could authors provide the appropriate reference. Only the authors name is mentioned.
Line 148: What is considered a high humidity?
Lines 159-162: This sentence is too long and therefore hard to interpret. Please adjust.
Line 163: “byexporuse” should be adjusted to “by exposure”.
Results
With editors permission, the heading of results should be adjusted to “Results and Discussion”, since the results are directly discussed.
Author Response
Authors aimed at assessing the potential of ssDNA functionalized single walled carbon nanotubes gas biosensor arrays for early detection of citrus trees infected with Huanglongbing disease. The subject is relevant for the maintenance of the yield of citrus. This manuscript is generally well written, roofreading is however required, since there are some minor grammatical revisions needed. I have outlined some suggestions for authors below.
Introduction
I would like to advise authors to make some adjustments to the structure of the introduction. Authors start by discussing the pitfalls of VOC analytical devices, however, the aim of the authors is to assess the ability of a specific eNose device for early detection of infected trees with HLB. I would like to advice authors to start by explaining the current situation regarding HLB and the need for novel detection methods. Thus, starting off with line 57 and the first two paragraphs can be transferred to line 92. This would increase reader comprehensibility, since all information regarding VOC analyses is then grouped together.
Response: The content has been edited according to the reviewer’ comment. Please check the introduction in the manuscript.
Line 45: “A primary challenge for gas sensors is therefore….” Here, “for gas sensors is therefore”, can cursiefbe removed from the sentence. Line 45: Could authors provide an appropriate reference for this statement?
Response: The wrong sentence has been corrected in the context. A primary challenge to design and develop robust, reliable gas sensors that are highly sensitive and selective to target gases, which can be operated at or close to room temperature under the influences of humidity.
“Neri, Giovanni. "First fifty years of chemoresistive gas sensors." Chemosensors 3.1 (2015): 1-20.”
Line 48: Reference 1 should be placed at the end of the sentence.
Response: The mistake has been revised.
Line 51: “bio/sensors” should be replaced by “bio-senors”
Response: The mistake has been revised.
Line 52: “Bio/receptors” should be replaced by “bio-receptors”
Response: The mistake has been revised.
Line 58: “Threathing” should be replaced by “threatening”.
Response: The mistake has been revised.
Line 60: Could authors provide an appropriate reference for this statement?
Response: Batool, A., et al. "Citrus greening disease-a major cause of citrus decline in the world-a review." Hort. Sci.(Prague) 34.4 (2007): 159-166.
Line 66: The names of bacteria should be displayed in italic. Please revise through whole manuscript.
Response:
Line 99: Between linalool and phenylacetaldehyde a comma is missing.
Response: The mistake has been revised.
Line 99: The names of the compounds should be checked for correct spelling.
Response: The mistake has been revised.
Line 112: “was” should be adjusted to “were”.
Response:
Materials and methods
Lines 132-134: did authors publish these gas sensing setup elsewhere? If so, please provide a reference. If not, could authors provide the precise settings.
Response: Badhulika, Sushmee, Nosang V. Myung, and Ashok Mulchandani. "Conducting polymer coated single-walled carbon nanotube gas sensors for the detection of volatile organic compounds." Talanta 123 (2014): 109-114.
Line 136: could authors provide the appropriate reference. Only the authors name is mentioned.
Response: Ramnani, Pankaj, Nuvia M. Saucedo, and Ashok Mulchandani. "Carbon nanomaterial-based electrochemical biosensors for label-free sensing of environmental pollutants." Chemosphere 143 (2016): 85-98.
Line 148: What is considered a high humidity?
Response: In the experiment, a cup of water was placed in the box to keep wet.
Lines 159-162: This sentence is too long and therefore hard to interpret. Please adjust.
Response: The sentence has been revised.
Line 163: “byexporuse” should be adjusted to “by exposure”.
Response: The mistake has been revised.
Results
With editors permission, the heading of results should be adjusted to “Results and Discussion”, since the results are directly discussed.

Reviewer 2 Report
This paper investigated the gas sensor which is based on the Single-stranded DNA functionalized SWNT. The detection of gas target is based on the partial charge induction on the SWNT by the adsorption of volatile organic compound (VOC) such as linalool. According to the reference paper [20], the detection of HLB is obtainable after a very sensitive detection and analysis of “fingerprint” of biomarkers VOCs. The authors introduced the concept of the gas sensor using SWNT, however, the gas sensor using the principle of FET has been widely studied.
I suggest the authors to revise the manuscript by enriching the discussion text and addressing the following points:
This paper should be more focused on what is the role of ss-DNA SWNT. I think that the rough sensing could be possible using SWNT only or simple FET. I couldn’t find why the ss-DNA functionalization on SWNT is important for the detection of VOC. The explanation how to integrate the sensitivity and selectivity to the ss-DNA SWNT is required. In figure 6, the S2DNA is much higher signal than S1DNA. Why is that? What is the role of the sequence of ssDNA? The relation between the sensing result of VOC and the detection of HLB is too weak. Figure 9A is missing. Figure 8 should be overlaid in 2D not 3D for better comparison. 3D is not necessary. Typo : in abstract, S2DNA is used without mentioning; Line 86, moreover ;
Author Response
This paper investigated the gas sensor which is based on the Single-stranded DNA functionalized SWNT. The detection of gas target is based on the partial charge induction on the SWNT by the adsorption of volatile organic compound (VOC) such as linalool. According to the reference paper [20], the detection of HLB is obtainable after a very sensitive detection and analysis of “fingerprint” of biomarkers VOCs. The authors introduced the concept of the gas sensor using SWNT, however, the gas sensor using the principle of FET has been widely studied.
I suggest the authors to revise the manuscript by enriching the discussion text and addressing the following points:
This paper should be more focused on what is the role of ssDNA SWNT. I think that the rough sensing could be possible using SWNT only or simple FET. I couldn’t find why the ss-DNA functionalization on SWNT is important for the detection of VOC. The explanation how to integrate the sensitivity and selectivity to the ss-DNA SWNT is required. In figure 6, the S2DNA is much higher signal than S1DNA. Why is that?
Response: For bare SWNTs, it is sensitive to most of VOCs, which cannot distinguish the type of VOCs. Here, we try to use s1DNA and s2DNA decorated the SWNTs to improve the sensitivity, and also change the selectivity for different VOCs that can take advantage of changes to establish the mathematic model. The types and numbers of bases in s1DNA and s2DNA are different, which will cause the different properties of s1DNA-SWNTs and s2DNA-SWNTs.
What is the role of the sequence of ssDNA? The relation between the sensing result of VOC and the detection of HLB is too weak.
Response: In this experiment, we try to monitor the changes of VOCs. Bare SWNTs can detect the VOCs, but it is hard to distinguish the kinds and concentrations of VOCs. Here, the two ssDNA are applied to modify the SWNTs because they are gas sensitive materials. So we want to establish a new biosensor array to solve the problem.
Figure 9A is missing.
Response: The Figure 9A has added in the manuscript.
Response:
Figure 8 should be overlaid in 2D not 3D for better comparison. 3D is not necessary.
Response: Thank you for your advice. The Figure has been revised.
Figure 8. Dynamic responses of different s2DNA-SWNTs towards different concentrations of phenylacetaldehyde vapors performed at VDS=0.1 V;
Typo : In abstract, S2DNA is used without mentioning; Line 86, moreover ;
Response: Thank for your advice. This sentence is not appropriate here, so that it has been deleted.

Reviewer 3 Report
Please see the attached document.

Author Response
The paper is related with the proposal of a new device to detect the Volatile Organic Compounds related with the HLB citrus disease. Without a doubt, there needs to be more research in order to get sensors to early detect HLB adequately. This paper is an important contribution in this way but some modifications and improvements are needed before it can be published in Sensors Journal:
The principal problem related to poor specificity and sensitivity described in the manuscript, prevents the reported device could be properly called a sensor. It is suggested rename the title and scope of study, because in its current state, the device is far from the state of art requirements of a true useful sensor. There should be main emphasis in the advantages of method.
It is required improve the specificity of method and it is mandatory to make additional studies related to possible interfering molecules and approaches to gain enough sensitivity for field measurements.
Response: Single-stranded DNA-Functionalized Single-Walled Carbon Nano-tubes Gas Biosensor Arrays for the Detection of Volatile Organic Compounds Biomarker released by the Huanglongbing Disease infected Citrus tree
It is suggested that edition changes must be done, some paragraph that deserve improving are listed below:
Line 55: “DNA molecules [9, 10] are naturally occurring polymers that have 55 many unique functions, including catalyzing chemical reactions [11] and controlling gene expression.”
This paragraph is out of context, additional argumentation must be inserted in order to connect DNA properties with the discussion above about the SWNT modification strategies.
Response: The introduction has been revised, please check the manuscript. This paragraph has is modified as following.
Line 120: “Two types of single-stranded DNA”
The authors must include additional justification about the selection criteria of DNA sequences utilized, in terms of sensitivity, specificity and their relation with detection mechanism or interaction with the VOC.
Response: Two types of single-stranded DNA were selected according to the previous work published by the Nicholas J. Kybert ---” Nano-Bio Hybrid Electronic Sensors for Chemical Detection and Disease Diagnostics”. Here, we try to use them to make a new biosensor array to diagnose the Huanglongbing infected citrus tree. The detection mechanism is discussed in the 3.4, but ssDNA-SWNTs can be considered as semiconductor material.
3.4 Sensing Mechanism
In order to experimentally identify the sensing mechanism, the ssDNA-SWNT chemiresistor devices were operated in a field-effect transistor mode using Si as the back-gate electrode. The changes in the transfer characteristic curves (IDS-VG, VDS = 0.1 V) upon functionalization with ssDNA and exposure to the VOCs was used to explain the electrostatic interactions of the DNA and VOC molecules with the SWNTs.
Fig. 9(a) shows transfer characteristics (IDS-VG) curves for s1DNA-SWNTs exposed to air and saturated phenylactaldehyde vapors. As illustrated in the figure, transconductance (slope of the transfer characteristics curve) changed after exposure to saturated phenylactaldehyde vapors. The increase in device transconductance upon exposure to analyte is attributed to change in local work-function of the metal contacts and band alignment when the VOC molecules are adsorbed on the metal (gold) electrodes. When the s2DNA-SWNT device was exposed to phenylalacetaldehyde vapors (Fig. 9(b)), the IDS-VG curve shifts to the negative direction, and threshold gate voltage (VTH) also decreases. This negative shift of VTH is attributed to adsorption of partially charged/polar VOC molecules that induce a screening charge (doping) of the SWNTs and shift the IDS-VG curve to a negative voltage. This mechanism, known as electrostatic gating, is commonly seen in most SWNT-based sensing devices.
Figure 9. Transfer characteristic (IDs-VG curves at VDS = 0.1 V) of (A) s1DNA-SWNTs and (B) s2DNA-SWNTs device in presence of air and saturated phenylactaldehyde.
Line 132: The rest of electrochemical measurements were collected using a gas sensing setup, which was designed and integrated by our group.
More information must be specified about the used setup related to electrochemical measurements and gas sensing procedures in order to justify and support a future fabrication of a true field portable sensor.
Response: The gas sensing setup has been used in “Ramnani, Pankaj, Nuvia M. Saucedo, and Ashok Mulchandani. "Carbon nanomaterial-based electrochemical biosensors for label-free sensing of environmental pollutants." Chemosphere 143 (2016): 85-98.”
Line 136 : SWNT based biosensor arrays were fabricated using the protocol reported by Ramnani et al.
The bibliographic reference of Ramnani protocol is missing, authors must include it in text and in the Reference Section.
Response: Badhulika, Sushmee, Nosang V. Myung, and Ashok Mulchandani. "Conducting polymer coated single-walled carbon nanotube gas sensors for the detection of volatile organic compounds." Talanta 123 (2014): 109-114.
Line 136: could authors provide the appropriate reference. Only the authors name is mentioned.
Line 161: “dome with a gas inlet and outlet ports for gas sealed glass dome with a gas inlet and outlet ports for gas was applied”
Correct the text repetition in this phrase
Response:
Line209: “As shown in Fig. 5(b), the threshold gate voltage (VTH) of bare SWNT was -12 V”
The Authors must check the value of threshold gate voltage of bare SWNT in this paragraph to be consistent with Figure 5 data.
Line309: “Based on these results, it is recommended that for real world applications water vapor must be removed during sampling of the VOCs from the citrus trees.”
The elimination of water vapor in natural sampling it is not a simple issue, otherwise the real use of reported sensor is challenged by a possible strategy to remove the vapor for a valid VOC analysis.
Response:
Line 322: “which were exhibited in Fig. 11.”
Figure 10 is missing! an improve in figures numbers must be done.
Response:
Line 348: “Supplementary Materials: The following are available online at www.mdpi.com/xxx/s1, Figure S1: title, Table S1: title, Video S1: title. “
The content of supplementary file is different of the description in this paragraph, please realize adequate change for consistency.
Supplementary Materials: The following are available online at www.mdpi.com/xxx/s1, Figure S1. IDS-VDS characteristics of SWNTs before and after functionalization with s1DNA at VG=0 V; Figure S2. Calibration curves of bare and ssDNA coated SWNTs sensor towards different concentrations of ethylhexanol vapors performed at VDS=0.1 V; ; Figure S3. Calibration curves of bare and ssDNA coated SWNTs sensor towards different concentrations of tetradecene vapors performed at VDS=0.1 V; Figure S4. Calibration curves of bare and ssDNA coated SWNTs sensor towards different concentrations of linalool vapors performed at VDS=0.1 V; Figure S5. Calibration curves of bare and ssDNA coated SWNTs sensor towards different concentrations of water vapors performed at VDS=0.1 V; Figure S7. PCA plot (PC1 vs. PC2) of scores using three sensors (bare SWNTs, s1DNA-SWNT and s2DNA-SWNT) for four VOCs and water teste; Figure S8. The real value versus the predicted value calculated by the NNF;Table S1. The concentrations of four VOCs at 25 ℃.

Round 2
Reviewer 2 Report
For the question " In figure 6, the S2DNA is much higher signal than S1DNA. Why is that?"
The Authors answered "The types and numbers of bases in s1DNA and s2DNA are different, which will cause the different properties of s1DNA-SWNTs and s2DNA-SWNTs."
However, the question is "How does the s2DNA behave differently from s1DNA, which results in a different outcomes? " The difference in gas sensing mechanism of s2DNA comparing the s1DNA should be discussed in the paper.
Author Response
For the question " In figure 6, the S2DNA is much higher signal than S1DNA. Why is that?"
The Authors answered "The types and numbers of bases in s1DNA and s2DNA are different, which will cause the different properties of s1DNA-SWNTs and s2DNA-SWNTs."
However, the question is "How does the s2DNA behave differently from s1DNA, which results in a different outcomes? " The difference in gas sensing mechanism of s2DNA comparing the s1DNA should be discussed in the paper.
Response: The sign and magnitude of ssDNA-SWNTs responses depended on both the DNA sequence used and the analyte. Nicholas and coworkers [43] proposed a possible reason that different ssDNA decorated with SWNTs, which will result in different complex, sequence-specific set of binding pockets located within a few nanometers of the SWNTs sidewall. Different VOC molecules were solvated by the DNA hydration layer and then bound in the pockets, resulting in different ssDNA-SWNTs responses.
Kybert, Nicholas J., et al. "Differentiation of complex vapor mixtures using versatile DNA–carbon nanotube chemical sensor arrays." ACS nano 7.3 (2013): 2800-2807.
